# Soil Silicon Amendment Increases *Phyllostachys praecox* Cold Tolerance in a Pot Experiment

**Zhuang Zhuang Qian [1,2], Shun Yao Zhuang [1,*], Qiang Li [1] and Ren Yi Gui [3]**

[1] State Key Lab of Soil and Sustainable Agriculture, Institute of Soil Science, Chinese Academy of Sciences, Nanjing 210008, China; zzqian@njfu.edu.cn (Z.Z.Q.); qli1074@gmail.com (Q.L.)

[2] College of Forestry, Nanjing Forestry University, Nanjing 210037, China

[3] State Key Lab of Sub-Tropical Sivlculture, Zhejiang A & F University, Hangzhou 311300, China; gry@zafu.edu.cn

[*] Correspondence: syzhuang@issas.ac.cn

**Abstract:** Cultivated bamboos are occasionally subjected to cold stress in winter, and silicon could improve their cold tolerance. However, evidence of the effect of Si on bamboos is still limited. Therefore, a batch and pot experiment was conducted for six months to investigate the effects of different Si fertilizer application rates (0, 0.5, 1.0, 2.0, 4.0, and 8.0 g kg$^{-1}$ of soil weight) on the physiological responses and photosynthesis parameters of *Phyllostachys praecox* under a simulated cold stress condition. The cold temperature was set to 5 °C, 0 °C, and −5 °C, successively. The bamboo biomass increased significantly when the Si amendment rate was at least 2.0 g kg$^{-1}$ ($P = 0.002$), and the highest biomass increase and root-to-canopy ratio were obtained with the 4.0 g kg$^{-1}$ Si amendment. Furthermore, the Si contents in all organs of the bamboos increased with the increase of the Si amendment rate. The highest content of Si among the other organs was observed in the leaf, and the content was 68.95 mg kg$^{-1}$ with the treatment of 4.0 g kg$^{-1}$. With the application of Si, the photosynthesis rate of bamboo leaves was significantly increased ($P = 0.008$). The Si-amended bamboo exhibited a cold tolerance that was associated with stimulating antioxidant systems, and the enzyme activities of superoxide dismutase, peroxidase, and catalase increased with the increase of the Si amendment rate, whereas the malondialdehyde content and cell membrane permeability decreased with all Si treatments. A low temperature of −5 °C exerted effects on the bamboo leaf chloroplasts, but the ultrastructures of the chloroplasts remained intact after Si treatment. These findings suggest that Si fertilizer enhances bamboo growth and the tolerance of bamboo plants to cold stress. However, a high application rate (8.0 g kg$^{-1}$) caused a decline in the bamboo biomass, compared to T4. Thus, a Si fertilization rate of 2.0~8.0 g kg$^{-1}$ is recommended for bamboos under cold conditions.

**Keywords:** bamboo forest; cold stress; physiological response; silicon fertilization

## 1. Introduction

Silicon is not a necessary element for higher plants, but it is essential for obtaining a high and sustainable yield for *Poaceae* crops. Si promotes the growth of various plants, especially those under abiotic and biotic stress conditions [1]. It is efficient in alleviating abiotic stresses, including salt stress, metal toxicity, drought stress, radiation damage, nutrient imbalance, high temperature, and freezing [2–9]. Previous studies have suggested that the following possible mechanisms underlie the Si-enhanced resistance of plants to abiotic stress: Stimulating antioxidant systems [10–13], reducing the transpirational bypass flow [14], reducing malondialdehyde (MDA), and improving root traits and the photosynthetic rate [15].



Cold stress is an abiotic stress that causes severe damage to membranes [16], and an increase in reactive oxygen species (ROS) in plants caused by freezing as well as increased lipid peroxidation, arising from the accumulation of ROS, are the major causes of membrane damage. MDA is a harmful lipid peroxidation product, which could reflect the extent of oxidative damage [17]. Under an ROS burst, superoxide dismutase (SOD) constitutes the first line of defense against ROS [18]. The activity of SOD has marked positive effects on the antioxidant capacity of plants. Peroxidase (POD) is generally considered to be a merely ROS-detoxifying enzyme [19]. Catalase (CAT) is a specific enzyme that catalyzes the dismutation of $H_2O_2$ into $O_2$ and $H_2O$, preventing the damaging effects of $H_2O_2$ accumulation and protecting cells from oxidative stress [20]. Silicon has been shown to ameliorate the damage of cold stress on plants. Liu et al. [21] found that Si addition increased antioxidant activities and decreased the MDA of *Cucumis sativus cv.* under chilling stress. Zhan et al. [22] also showed that Si amendment alleviates chilling stress in *D. brandisii* plantlets, exhibiting increased CAT and SOD activities and decreased MDA. Moreover, Si forms deposits and undergoes polymerization, forming "phytoliths" in cells and intercellular spaces [23,24]. The strength and rigidity of the tissues improves [25], and thus cold tolerance is enhanced. Several researchers concluded that Si not only acts as a physical or mechanical barrier in plants but is also involved in metabolic and physiological activities [26].

Despite the wealth of information on the beneficial effects of Si on plants, the mechanisms underlying Si-mediated alleviation under freeze stress remain poorly understood. *Phyllostachys praecox* is among the bamboos with edible shoots in the region of Southeast China. This bamboo species has a high yield and plays an important role in the local economy. Accordingly, its sustainable production is essential for farmers. However, it is subjected to low temperatures and freeze stress in winter and often shows poor growth. Zhou et al. [27] reported that the ice damage rate of typical subtropical forests varied between 25% and 81%. A previous study found that the shoot output decreased by 32% per hectare in 2008 [28], and the bamboo freeze injury index of Lin-an City was 51% in 2016 [29]. Thus, measures to mitigate low-temperature stress are beneficial for bamboo production. Graminaceous plants absorb much more Si than other species [30]. Similar to many *Poaceae* species, bamboos accumulate Si, which may alleviate stress at low temperatures and freeze stress. However, Si uptake and accumulation in *P. praecox*, and the effect of Si on the resistance of *P. praecox* to low temperatures, have not been studied. Therefore, this study aimed to (1) investigate how Si amendment to *P. praecox* affects Si accumulation, (2) elucidate the relationship between Si content and *P. praecox* growth, and (3) explore the mechanism underlying the enhanced resistance of plants to low temperatures, after Si addition, by performing a plant physiological indicator analysis.

## 2. Materials and Methods

### 2.1. Bamboo and Soil

*P. praecox* is a bamboo that is mainly cultivated in Lin-an City (30°16′24.58″ N, 119°35′14.59″ E), with intensive management. Bamboos usually grow from seeds, underground rhizomes, or *P. praecox* flowers in the study site, but they do not form seeds, and they extend through the underground rhizome. Therefore, we selected bamboo rhizome for our pot experiment. The length of rhizomes was approximately 200 mm, their diameter was approximately 10 mm, and their weight was approximately 300 g. The rhizome sprouts of *P. praecox* were excavated in 2015 from the bamboo field and cultured in a greenhouse. Before the pot experiment, rhizome sprouts with similar sizes and weights were incubated.

Soil (0–25 cm) in the pot experiment was collected from the bamboo garden in Zhejiang Agriculture and Forestry, Lin-an City, ZheJiang Province, China. The soil type was classified as Ultisol. The soil basic physicochemical properties were as follows: pH, 5.31; available Si, 43.51 mg kg$^{-1}$; soil organic matter, 16.8 g kg$^{-1}$; soil total nitrogen, 769 mg kg$^{-1}$; available potassium, 25 mg kg$^{-1}$; hydrolyzed nitrogen, 81.62 mg kg$^{-1}$; and available phosphorus, 68.5 mg kg$^{-1}$. The soil sample was air-dried, passed through a 2 mm mesh sieve, and, for the improvement of the soil structure, sieved soils were mixed with perlite (bamboo rhizomes prefer soil that is loose and permeable and are easily damaged

under hypoxia conditions, so perlite can increase the permeability of soil, which is beneficial for the growth of bamboo) in a 3:1 ratio (volume volume$^{-1}$) for the pot experiment.

## 2.2. Bamboo Pot Experiment

The bamboo pot experiment was conducted in 2016 in the greenhouse in Zhejiang Agriculture and Forestry, China. The light transmittance rate in the greenhouse was 88%, and the temperature ranged from 20 to 28 °C, which was suitable for bamboo growth. The height of the experimental plastic pots was 250 mm, and the diameter of the pots was 300 mm. Each pot had five holes for drainage in the bottom and was filled with 3 kg of soil. The experiment involved six treatments, with Si fertilizer application rates of 0, 0.5, 1.0, 2.0, 4.0, or 8.0 g kg$^{-1}$, marked as T0, T0.5, T1, T2, T4, and T8, respectively. Each treatment had 20 pots, and four bamboos were planted in each pot. The photosynthesis parameters (photosynthesis rate, water use efficiency, $CO_2$ of intercellular space, and stomatal conductance) of the bamboo leaves were measured every month. After six months, five bamboo pots from each treatment were randomly selected for biomass measurement. The root, rhizome, leaf, and stem of the bamboo plant were collected for further analysis. The root-to-canopy ratio was calculated by the underground biomass and overground biomass, and the equation was:

$$\text{root} - \text{to} - \text{canopy ratio} = \frac{\text{underground biomass}}{\text{overground biomass}}. \tag{1}$$

Low Temperature Incubation

Three bamboo pots (bamboos in pots were basic, with the same height and ground diameter) were chosen randomly from every treatment for further incubation. The three bamboo pots from each treatment were divided into three culture boxes, and the temperature of the three culture boxes was set to 5, 0, and −5 °C. A total of 18 bamboo pots were cultured for three days under the specified temperatures. After the three types of temperature treatments (5, 0 and −5 °C), the functional leaves of bamboo were collected for plant physiological analysis.

## 2.3. Si in Plant Measurement

The excavated bamboo was washed and divided into root, leaf, stem, and rhizome. It was then soaked in 0.5 M HCl for 20 s, followed by three to four rinses in distilled water, and then dried to stable weight at 65 °C. After being dried, the bamboo samples were ground and passed through a 0.25 mm mesh sieve for Si measurement. The samples were then microwave digested in a mixture of 3 mL of 62% (w w$^{-1}$) $HNO_3$, 3 mL of 30% (w w$^{-1}$) hydrogen peroxide, and 2 mL of 46% (w w$^{-1}$) hydrofluoric acid (HF), and the digested sample was diluted to 100 mL with 4% (w v$^{-1}$) boric acid. The Si concentration in the digested solution was determined by the colorimetric molybdenum blue method at 600 nm [31].

## 2.4. Photosynthesis Parameter Measurement and Physiological Indicator Analysis

For the photosynthesis parameter measurement, three sunlight-exposed leaves of each bamboo plant were measured, two bamboo plants were measured in a single pot, and three pots were measured per treatment. The photosynthesis rate (P$n$, µmol $CO_2 \bullet$ m$^{-2}$ s$^{-1}$), transpiration rate (T$r$, mmol $H_2O \bullet$ m$^{-2}$ s$^{-1}$), water use efficiency (WUE, µmol $CO_2 \bullet$ mmol$^{-1}$ $H_2O$), $CO_2$ of intercellular space (C$i$, µL$\bullet$L$^{-1}$), and stomatal conductance (G$s$, mmol$\bullet$m$^{-2}$ s$^{-1}$) of the bamboo functional leaves were measured using a GFS-3000 (WALZ, Effeltrich, Germany). During the measurement, the environmental condition was set at a sunlight intensity of 1300 µmol$\bullet$m$^{-2}\bullet$s$^{-1}$, ambient $CO_2$ concentration of 400 ± 20 µmol$\bullet$m$^{-2}\bullet$s$^{-1}$, and leaf temperature of 25 ± 1 °C [32].

After three days of low-temperature incubations at 5, 0, and −5 °C, the SOD, POD, CAT, MDA, and cell membrane permeability (CMP) activities of the leaves were measured by the nitroblue

tetrazolium reduction [33], guaiacol colorimetric [34], ultraviolet absorption [35], thiobarbituric acid [36], and electric conductivity methods, respectively [37].

## 2.5. Observation of Bamboo Leaf Chloroplast Ultrastructure

Mature bamboo leaves of the T0 and T4 treatments were collected from the −5 °C culture boxes and cut into small pieces (1 mm × 1 mm). The T0 and T4 treatments were chosen, because T0 had no Si amendment, and T4 showed the highest leaf biomass among all treatments. We hypothesized that T4 could exhibit a better visual expression than the other treatments. The chosen section was located in the middle of the leaf, at 1 cm from the main vein. Small pieces of these leaf materials were immersed in 4% (m m$^{-1}$) glutaraldehyde overnight at 4 °C, thoroughly rinsed with 0.1 M phosphate buffer (pH 7.0), and fixed with 1% (m m$^{-1}$) osmium acid for 14 h at −4 °C. Then, the samples were sequentially immersed in 50% alcohol for 15 min, 70% alcohol for 15 min, 80% alcohol for 15 min (2 times), 90% alcohol for 15 min (2 times), 95% alcohol for 15 min (2 times), 100% alcohol for 10 min (2 times), and 100% acetone for 10 min (2 times). Finally, the samples were embedded in molds with epoxy resin (Epson 812) at 60 °C for 24 h. Ultrathin sections were stained by uranyl acetate and lead citrate and observed under a microscope (XSP-8CA, Shanghai, China). Cell photomicrographs were taken by transmission electron microscopy (JEM-1200EX, Tokyo, Japan) [38].

## 2.6. Data Analysis

The study was carried out in a completely randomized design. The SOD, POD, and CAT activity, CMP and MDA concentration, photosynthesis parameters (photosynthesis rate, water use efficiency, $CO_2$ of intercellular space, and stomatal conductance) in the plant leaves, biomass, and root-to-canopy ratio were examined statistically by an analysis of the variance and means of three replicates were subjected to Duncan's test at a 5% probability level using IBM SPSS Statistics 20.0 (SPSS Inc., Chicago, IL, USA).

Furthermore, a multiple regression analysis was adopted. The relationship between the Si content in the plant and the amended rate can be described by the following equation:

$$Y = ax^2 + bx + c \tag{2}$$

where $Y$ is the Si content in bamboo, $x$ is the Si content amended in the pot, and $a$, $b$, and $c$ are the parameters for the equation.

The relationship between the Si amendment and enzyme content under −5 °C treatment could also be described by Equation (2), where $Y$ is the SOD, POD, and CAT activity and MDA concentration in the bamboo leaves, and $x$ is the Si content amended in the pot. Correlation coefficients and multiple regression coefficients were calculated using Microsoft Excel.

## 3. Results

### 3.1. Bamboo Biomass Change with Si Amendment

The bamboo biomass increased significantly when the Si amendment rate was above T2, after six months of incubation (Figure 1) ($P = 0.002$). However, no significant difference was found in the T2, T4, and T8 treatments ($P > 0.05$). The high rate of Si amendment (T8) led to a slight decline in biomass, compared to T4. Similarly, the root-to-canopy ratio showed the same trend as the biomass ratio increased. T4 had the highest biomass increase and root-to-canopy ratio among the treatments. These results indicated that Si amendment improves bamboo growth under low temperatures.

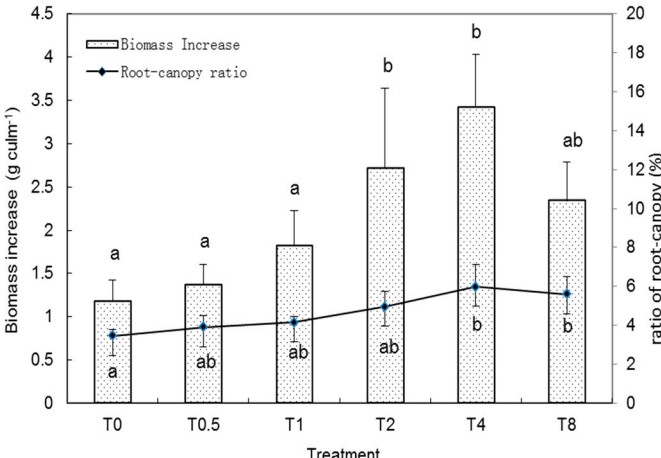

**Figure 1.** Bamboo biomass change after Si amendment. Values followed by the same letter(s) are not significantly different at *P* < 0.05, according to Duncan's multiple range tests. The measurements were taken after six months of Si application at room temperature.

### 3.2. Si Content in Bamboo Plants

The Si contents in the leaf, stem, rhizome, and root of the bamboos increased with the increase of the amendment rate (Table 1). In the leaves, the highest Si content was 68.95 mg kg$^{-1}$, which occurred in T4. In the stem, rhizome, and root, the highest Si contents were all observed in T8. Relatively, the Si content in the leaf was higher than in the stem, rhizome, or root. The Si content in the bamboo plant body can be arranged as follows: leaf > stem > rhizome > root. The results also showed that the total Si content in the bamboo was closely related to the Si amendment rate. Table 2 shows that the Si content in all bamboo organs could be well fitted by the equation, and the equations for the leaf, stem, rhizome, and root were $y = -1.2901x^2 + 15.311x + 26.607$, $y = -0.3743x^2 + 4.5539x + 10.923$, $y = -0.1309x^2 + 1.5841x + 3.2716$, and $-0.1003x^2 + 1.5574x + 3.1215$, respectively. All $R^2$ values were above 0.95. The maximum values of the leaf, stem, rhizome, and root biomass were obtained at 5.93, 6.08, 6.05, and 7.76 mg kg$^{-1}$, respectively. This result suggests that bamboo had a maximal Si uptake, regardless of the Si amendment rate.

**Table 1.** Si content in bamboo parts with various Si treatments (mg kg$^{-1}$). Data are the means ± SD of three replicates. Different letters in the same column indicate significant differences, based on Duncan's multiple range tests at the 0.05 level. The measurements were taken after six months of Si application at room temperature.

| Treatment | Leaf | Stem | Rhizome | Root |
|-----------|------|------|---------|------|
| T0 | 27.77 ± 4.70 a[1] | 11.18 ± 1.53 a | 3.50 ± 0.72 a | 3.07 ± 0.99 a |
| T0.5 | 31.93 ± 3.68 a | 11.35 ± 1.40 a | 3.31 ± 0.81 a | 3.69 ± 1.18 a |
| T1 | 43.19 ± 3.25 b | 16.71 ± 4.37 ab | 4.82 ± 0.95 a | 5.33 ± 0.71 b |
| T2 | 48.88 ± 2.13 b | 18.88 ± 3.69 b | 5.64 ± 1.10 ab | 6.14 ± 0.74 bc |
| T4 | 68.95 ± 10.11 c | 22.60 ± 3.71 c | 7.92 ± 2.51 bc | 7.16 ± 1.11 c |
| T8 | 66.26 ± 9.20 c | 23.50 ± 5.57 c | 9.13 ± 3.25 c | 7.63 ± 1.19 c |

**Table 2.** Parameters for the equation for the Si content in the bamboo plant and the Si-amended plant. Significant differences were based on Duncan's multiple range tests at the 0.05 level.

| | *a* | *b* | *c* | $R^2$ | *P* |
|---|-----|-----|-----|-------|-----|
| Leaf | −1.290 | 15.31 | 26.61 | 0.9828 | 0.003 |
| Stem | −0.3743 | 4.554 | 10.92 | 0.9566 | 0.009 |
| Rhizome | −0.1003 | 1.557 | 3.122 | 0.9789 | 0.003 |
| Root | −0.1309 | 1.584 | 3.272 | 0.9585 | 0.008 |

### 3.3. Photosynthesis Parameters

As shown in Table 3, with Si amendment, the photosynthesis rate ($Pn$) of the bamboo leaves increased linearly, from 7.03 μmol $CO_2 \bullet$ m$^{-2}$ s$^{-1}$ to 11.89 μmol $CO_2 \bullet$ m$^{-2}$ s$^{-1}$, and $Pn$ increased significantly when the Si amendment rate was above 4.0 g kg$^{-1}$ ($P = 0.008$). However, the bamboo transpiration rate (Tr) was significantly reduced, from 3.74 mmol $H_2O \bullet$ m$^{-2}$ s$^{-1}$ to 1.93 mmol $H_2O \bullet$ m$^{-2}$ s$^{-1}$ ($P = 0.001$). The WUE increased from 2.80 μmol $CO_2 \bullet$ mmol$^{-1}$ $H_2O$ to 7.39 μmol $CO_2 \bullet$ mmol$^{-1}$ $H_2O$, and it increased significantly when the Si amendment rate was above 4.0 g kg$^{-1}$ ($P = 0.01$). The $CO_2$ concentration of the intercellular space and stomatal conductance showed no significant difference with all the Si amending treatments ($P > 0.05$).

**Table 3.** Photosynthesis parameters of the bamboo leaf with various treatments. Data are the means ± SD of three replicates. Different letters in the same column indicate significant differences, based on Duncan's multiple range tests at the 0.05 level. The measurements were taken after six months of Si application at approximately 25 °C.

| Treatment | Photosynthesis Rate ($Pn$) (μmol $CO_2 \bullet$ m$^{-2}$ s$^{-1}$) | Transpiration Rate ($Tr$) (mmol $H_2O \bullet$ m$^{-2}$ s$^{-1}$) | Water Use Efficiency (WUE) (μmol $CO_2 \bullet$ mmol$^{-1}$ $H_2O$) | $CO_2$ of Intercellular Space ($Ci$) (μL L$^{-1}$) | Stomatal Conductance ($G_s$) (mmol $\bullet$ m$^{-2}$ s$^{-1}$) |
|---|---|---|---|---|---|
| T0 | 7.03 ± 2.69 a | 3.74 ± 1.26 a | 2.80 ± 1.83 a | 224.33 ± 45.89 a | 109.0 ± 14.91 a |
| T0.5 | 7.01 ± 0.65 a | 2.64 ± 0.89 b | 2.38 ± 0.58 ab | 240.62 ± 40.19 a | 99.4 ± 28.6 a |
| T1 | 8.11 ± 3.97 ab | 2.5 ± 0.32 b | 3.84 ± 1.58 ab | 241.12 ± 45.21 a | 97.63 ± 12.83 a |
| T2 | 9.57 ± 2.6 abc | 2.58 ± 0.89 b | 3.94 ± 1.17 ab | 240.19 ± 47.57 a | 111.07 ± 30.65 a |
| T4 | 10.11 ± 2.55 bc | 2.27 ± 0.6 b | 4.55 ± 1.12 b | 228.47 ± 19.73 a | 103.96 ± 17.01 a |
| T8 | 11.89 ± 1.63 c | 1.96 ± 0.75 b | 7.39 ± 3.79 c | 230.99 ± 37.23 a | 118.7 ± 38.06 a |

### 3.4. Physiological Indicators Treated with Low Temperature

Figure 2 shows that the SOD activity was significantly higher at 0 °C than at 5 °C and −5 °C ($P = 0.001$). With Si amendment, the SOD activity increased with all temperature treatments and was in the order of T8 > T4 > T2 > T1 > T0.5. As shown in Figure 3, the low temperature of −5 °C significantly reduced the POD activity of the bamboo leaves ($P = 0.001$). The Si amendment increased the POD activity of leaves at various temperatures, and no significant difference was found at 5 °C ($P > 0.05$). At 0 °C and −5 °C, the POD significantly increased with the increasing rate ($P = 0.007$, $P = 0.002$). Similarly, with no more than 8 g kg$^{-1}$ application rates, the SOD and CAT activity of the bamboo leaves increased with the increasing Si amendment (Table 4). The highest CAT activity was recorded at −5 °C with all Si treatments (Figure 4). Unlike the SOD and CAT, the Si amendment decreased the MDA concentration (Figure 5). The highest MDA concentration was recorded at −5 °C. At 0 °C and −5 °C, the MDA significantly decreased with the increasing rate ($P = 0.01$, $P = 0.002$) (Table 4). The CMP of the bamboo leaves increased with the decreasing temperature (Figure 6). The Si amendment could reduce the CMP at various temperatures, and the CMP decreased significantly when the Si amendment rate was above 2.0 g kg$^{-1}$ at −5 °C (Figure 1) ($P = 0.001$).

**Table 4.** Parameters for the equation for the Si amendment and enzyme content at −5 °C. Significant differences were based on Duncan's multiple range tests at the 0.05 level. SOD, superoxide dismutase; POD, peroxidase; CAT, catalase; MDA, malondialdehyde.

| | $a$ | $b$ | $c$ | $R^2$ | $P$ |
|---|---|---|---|---|---|
| SOD | −1.783 | 23.266 | 633.6 | 0.89389 | 0.016 |
| POD | −0.7196 | 9.935 | 112.1 | 0.96943 | 0.002 |
| CAT | −0.3301 | 6.390 | 27.49 | 0.98354 | $9.82 \times 10^{-4}$ |
| MDA | 0.05001 | −1.202 | 27.95 | 0.97371 | 0.002 |

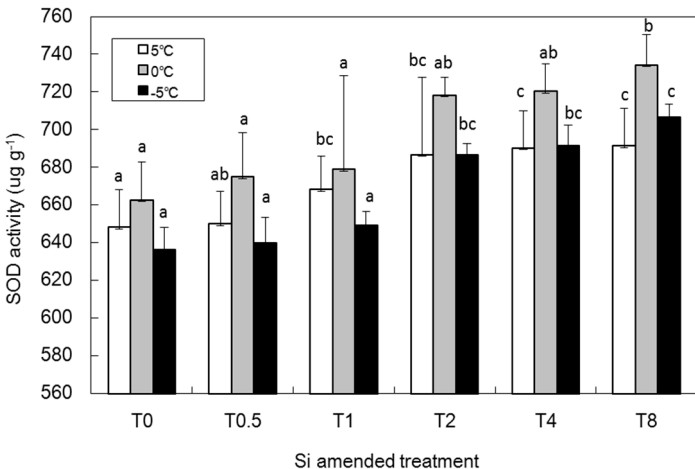

**Figure 2.** Effect of Si amendment on the SOD activity of leaves at various temperatures. Data are the means ± SD of three replicates. Different letters at the same temperature indicate significant differences, based on Duncan's multiple range tests at the 0.05 level (the same hereinafter). SOD, superoxide dismutase. SOD activities were measured after three days of low-temperature incubations at 5 °C, 0 °C, and −5 °C (the same hereinafter).

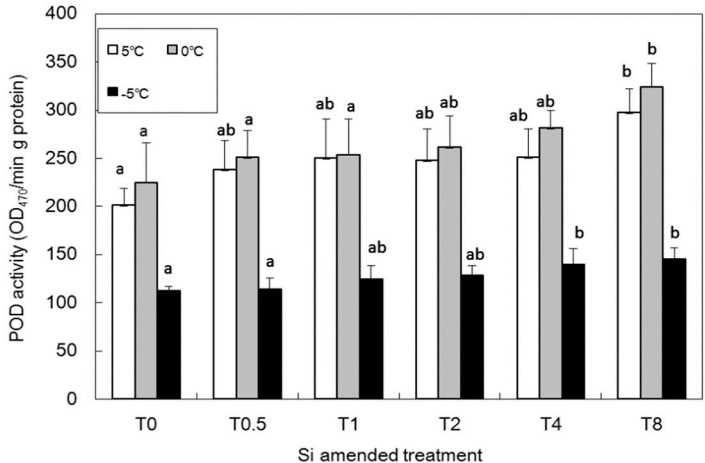

**Figure 3.** Effect of Si amendment on the POD activity of leaves at various temperatures. POD, peroxidase.

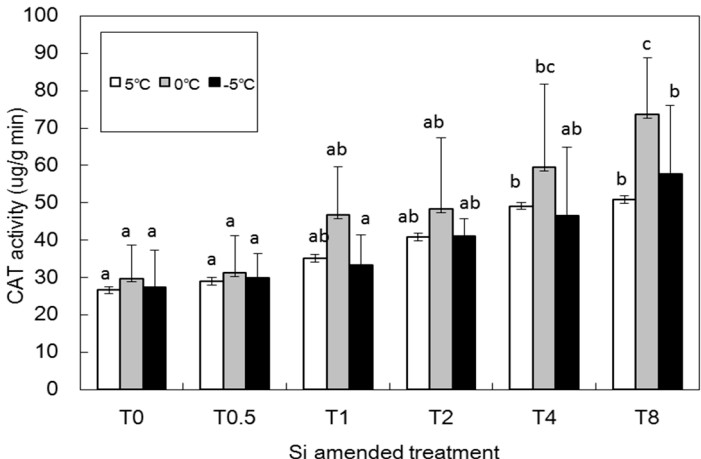

**Figure 4.** Effect of Si amendment on the CAT activity of leaves at various temperatures. CAT, catalase.

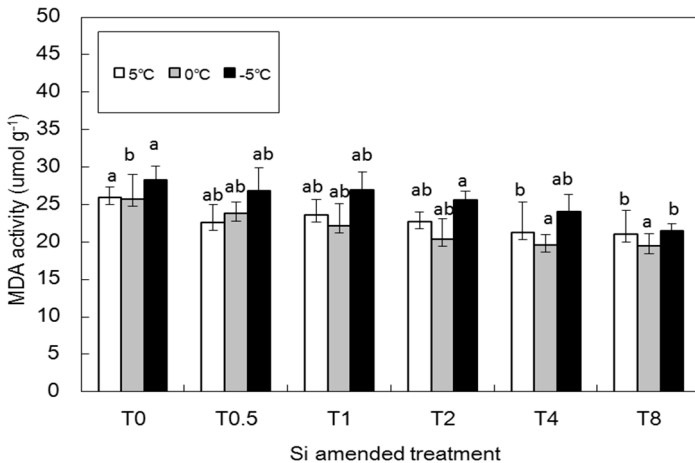

**Figure 5.** Effect of Si amendment on the MDA concentration of leaves at various temperatures. MDA, malondialdehyde.

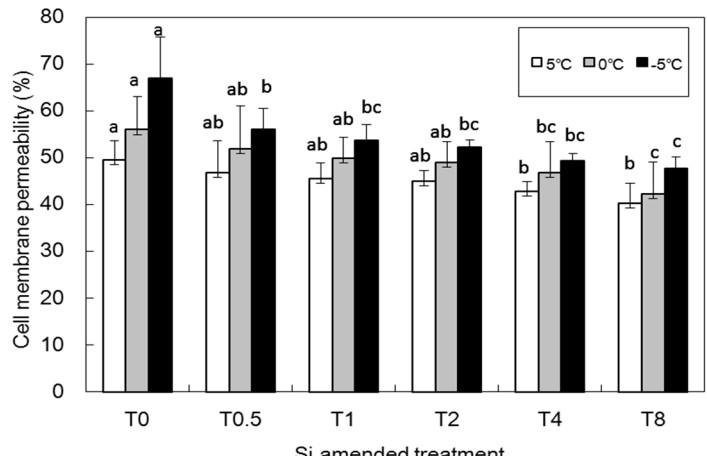

**Figure 6.** Effect of Si amendment on the cell membrane permeability of leaves at various temperatures.

*3.5. Effect of Si Amendment on Bamboo Leaf Chloroplast Ultrastructure*

Figure 7 shows that the low temperature of −5 °C exerted a significant effect on the bamboo leaf chloroplast. The chloroplast swelled to a circular shape and separated with the cell membrane. Even worse, the chloroplast membrane ruptured, and the grana disintegrated, while some dissolved. The osmiophillic number increased, and some small vesicles were found in the cell matrix (Figure 7A,B). The chloroplast showed an intact ultrastructure with Si treatment (Figure 7C,D) and came in close contact with the cell membrane. The membrane of the chloroplast was distinct and full. The application of Si prevented low-temperature stress in the bamboo leaf chloroplast and decreased the degree of damage affecting bamboo growth.

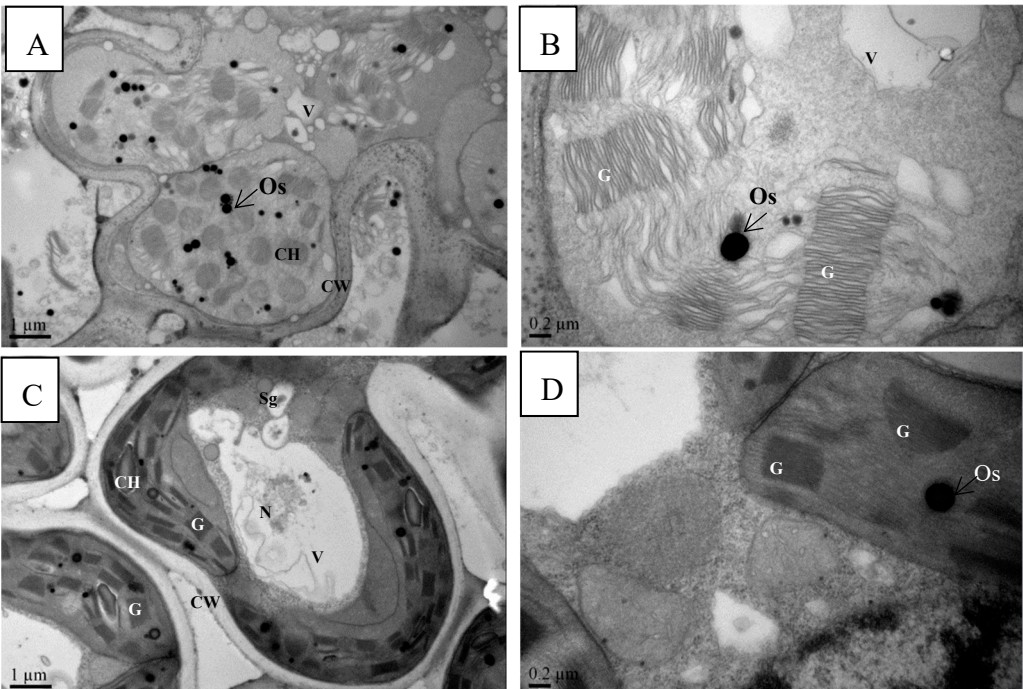

**Figure 7.** Effect of Si amendment on the chloroplast ultrastructure of bamboo leaves at −5 °C (**A**) 15,000×, chloroplast of control; (**B**) 40,000×, chloroplast of control; (**C**) 15,000×, chloroplast of Si treatment; (**D**) 40,000×, chloroplast of Si treatment). Os: Osmiophile globule; Cw: cell wall; V: vacuole; Ch: chloroplast; G: Granum; N: nucleus; Sg: starch grain. The ultrastructure of bamboos leaves (T0 and T4) were observed after three days of −5 °C temperature incubations.

## 4. Discussion

Temperature determines the distribution of bamboo [28,39,40], and *P. praecox* is usually subjected to low temperatures. Thus, Si amendment has been considered one of the best methods to mitigate low-temperature stress in *P. praecox*, because Si is beneficial for bamboo growth [41,42].

The results of this study indicate that Si amendment significantly improves the bamboo root-to-canopy ratio ($P = 5 \times 10^{-4}$), which indicate that the growth of underground organs increased, such that bamboo roots easily absorbed nutrients and water from the soil. Moreover, Si increased the total biomass of bamboo and especially accumulated in bamboo leaves, which was beneficial for the bamboo growth and enhanced the cold tolerance of bamboo. In our findings, Si significantly increased the photosynthesis rate of bamboo leaves ($P = 0.008$) and the ultrastructures of the chloroplasts, which was consistent with Detmann et al. [43]. Moreover, Si improved the leaf thickness and surface area of chloroplasts. In addition, Si improved the erectness of leaves, so as to indirectly improve the whole-plant photosynthesis [44]. The WUE increased significantly with Si amendment ($P = 0.01$) due to increased photosynthesis and reduced water Tr. The stomatal conductance and mesophyll conductance were limitations to the photosynthetic capacity of leaves [45], but there was no significant difference of stomatal conductance in all the Si amending treatments ($P > 0.05$), which suggested that stomatal produce posed a less significant limitation on photosynthesis. However, the high application rate of Si fertilization may have adverse effects on bamboo growth. Active and passive Si uptake both existed in plants, and in the passive process, the supply of silicon in the bamboo depends on the availability of $Si(OH)_4$ in the soil of their growth area and the rates of water uptake and evaporation [42]. When too much silicon accumulated in the bamboo, the transpiration rate significantly decreased ($P = 0.001$), and the passive process decreased, which caused the decrease in biomass. Thus, Si fertilization is an effective measure for improving the tolerance of bamboos to cold stress, and a Si fertilization rate of 2.0~8.0 g kg$^{-1}$ is recommended for bamboo growth.

The increased growth was closely related to the stimulation in the enzymatic antioxidant system under cold stress. ROS, including superoxide ($O_2 \bullet -$), $H_2O_2$, and hydroxyl radicals (OH•), were generated in the plants due to environmental stresses [46,47]. Chloroplasts are the major sites for ROS generation, and oxygen generated in the chloroplasts during photosynthesis can accept electrons passing through the photosystems, thus forming $O_2 \bullet -$ [48]. When plants were exposed to cold stress factors, an excessive accumulation of ROS could result in oxidative damage to plant tissues [49–51]. Moradtalab et al. [52] suggested that the major effect of Si on improving the cold tolerance of plants is the mitigation of oxidative stress. Therefore, cold stress resistance is associated with an enhanced antioxidative defense system, which includes antioxidant compounds and several antioxidative enzymes. In our study, Si application stimulated the SOD, POD, and CAT activity of bamboo leaves at various temperatures. Moreover, the content of SOD, POD, and CAT activity increased with the increase of the Si application rate. SOD is the major $O_2 \bullet -$ scavenger, and its enzymatic action results in $H_2O_2$ and $O_2$ formation. Since $H_2O_2$ can diffuse directly across the membrane, $H_2O_2$ produced in chloroplasts, mitochondria, and other organelles can also diffuse into the peroxisomes and be scavenged by CAT [53]. The lowest POD content was observed at −5 °C. A low content of $H_2O_2$ in plant cells is removed by POD. When the $H_2O_2$ content is very high, CAT is mainly responsible for its removal [54]. Temperature stress results in an increase in the MDA content [55]. Therefore, a low temperature of −5 °C increased the MDA content of bamboo leaves, and the decreased MDA and CMP content indicated that the stability of the cell membrane was improved, the CMP was decreased, and the membrane lipid peroxidation was alleviated, induced by the low temperature, which enhanced the cold tolerance of the bamboos. Plants have developed an antioxidant system, including CAT, guaiacol peroxidase (GPX), glutathione peroxidase (GSH-Px), and the ascorbate–glutathione cycle, to scavenge ROS. The ascorbate–glutathione cycle, including glutathione (GSH), ascorbate (AsA), and related enzymes, such as glutathione reductase (GR) and dehydroascorbate reductase (DHAR), is an important way to scavenge the toxic products in chloroplasts and other non-photosynthetic tissues [56]. Our study did not examine the dynamic changes of GR and DHAR, but showed that the antioxidant enzymes of SOD, POD, and CAT were stimulated by Si addition under cold conditions. In addition, under a low temperature, the morphological structures of organelles change, along with the damaged chloroplasts and starch granules [57]. In our study, the shape of chloroplasts was improved by Si amendment, and the chloroplast membranes were stabilized. Our results supported previous studies that found that, under cold stress, Si amendment enhanced oxidative defense systems and the chlorophyll content, and reduced the absolute electrolyte leakage quantity of corn plantlets [21,22,58,59]. These results indicate that Si enhances the resistance of plants to cold stress. In addition, further field studies should be conducted to test the effects of silicon in improving bamboo cold tolerance. Our low temperature treatment was not conducted for as long as the field situation required, and our study of the mechanism of Si-mediated cold stress needs further development.

The mechanism of Si in improving bamboo cold tolerance was as follows: (1) Si improves the bamboo biomass, the Si content in bamboo leaves, the chlorophyll content, and the ultrastructure of chloroplast, which was beneficial in increasing the photosynthesis rate. (2) Antioxidant systems are stimulated by Si, and play an important role in alleviating the peroxidation damage induced by cold stress.

## 5. Conclusions

Our results show that Si application improved the growth of *P. praecox* and increased the biomass, root-to-canopy ratio, and P*n* of bamboo leaves. The method of soil Si amendment was effective in alleviating cold stress in *P. praecox*. It resulted in an improved chloroplast and cell structure, increased CAT, POD, and SOD activities, and decreased MDA and CMP levels. These findings suggest that Si is beneficial for bamboo growth, although a high rate of application (8 g kg$^{-1}$) may have adverse effects on bamboo growth, so we recommended a Si fertilization rate range of 2.0~8.0 g kg$^{-1}$, and Si fertilizer enhances the tolerance of bamboo plants to cold stress. Accordingly, Si fertilization is recommended

for bamboos under cold conditions. Furthermore, more field studies need to be conducted to test the practical application of these findings.

**Author Contributions:** Conceptualization, Z.Z.Q., S.Y.Z., Q.L., and R.Y.G.; Methodology, Z.Z.Q., S.Y.Z., Q.L., and R.Y.G.; Validation, Z.Z.Q., S.Y.Z., Q.L., and R.Y.G.; Formal Analysis, S.Y.Z.; Resources, S.Y.Z. and R.Y.G.; Visualization, Z.Z.Q., S.Y.Z., Q.L., and R.Y.G.; Investigation, Z.Z.Q. and Q.L.; Resources, S.Y.Z. and R.Y.G.; Software, Z.Z.Q. and Q.L.; Data Curation, Z.Z.Q. and Q.L.; Writing—Original Draft Preparation, Z.Z.Q. and S.Y.Z.; Writing—Review and Editing, Z.Z.Q. and S.Y.Z.; Supervision, S.Y.Z. and R.Y.G.; Project Administration, S.Y.Z.; Funding Acquisition, S.Y.Z. and R.Y.G.

**Funding:** The authors are grateful for financial support from the National Natural Science Foundation of China (41671296), National Key R & D Project of China (2016FYE0112700), and Science and Technology Department of Zhejiang Province of China (2017C02016).

**Acknowledgments:** We are grateful for experimental support from Yuhe Zhang and Xin Wang.

**Conflicts of Interest:** The authors declare no conflict of interest.

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
