# Peer review of "Soil Silicon Amendment Increases Phyllostachys praecox Cold Tolerance in a Pot Experiment"

_forests, doi:10.3390/f10050405_

Round 1

Reviewer 1 Report

GENERAL COMMENTS

This study provides interesting and useful information regarding the effects of Si amendment on bamboo productivity and cold tolerance. A pot experiment was used to test 6 different Si application rates, including the control (no application). Numerous parameters were measured and tested that shed light on the effects of Si application and the potential mechanisms driving the observed changes. My main concern with the manuscript is that the methods are not clear, and thus the results, interpretation of the results, and conclusions are also unclear. Specifically, it is unclear exactly which measurements were taken after the 6-month experiment (20-28°C) or the 9-day cold-stress conditions (5, 0, -5°C), and it seems that results that were not measured after the cold-stress conditions are conflated with results that were. Main results and conclusions not related to the cold conditions part of the experiment should be made abundantly clear (but are not necessarily less important). In general, the writing clarity and organization need extensive revisions (see below). Additionally, there were numerous grammatical errors throughout the manuscript (which I have not commented on) that should be addressed. My specific comments are below.

SPECIFIC COMMENTS

Title:

-The title would be more effective if it reflects the main results/conclusion of the study

Abstract:

-State temperature of 6-month growing period; clarify how the experiment transitioned to the cold-stress experiment, the length of this part of the experiment, and when and what reported measurements were taken; specify and clearly define what is being referred to by “cold conditions” under which the plants experience cold stress

-There are generally too may abbreviations in this manuscript; please limit them as much as possible

-L18: “at least” 2.0 g/kg, not “more than”

-L19: remove sentence starting wtih “A slight decline…” as it’s not needed and is confusing

-Conclusion: provide recommended range of Si amendment rate

Introduction:

-Check to make sure that all abbreviations are spelled out in full the first time

-Again, please limit abbreviations used in the manuscript

-L40-52: this is just a list; please limit the discussion to only what is relevant to your study and, more specifically, what you will actually look at; make connections clear, rather than providing this in list format (that is, what are these features you will measure and why are they important)

-L59-63: this portion belongs at the beginning of the introduction section; provide more information about why this particular study is needed; provide citations of previous work

-L66: (DM) ???

Methods:

-L74-77: belongs in introduction section

-L86: were rhizome seedlings for each treatment randomly selected? Can you provide the range of starting size and weight of rhizome seedlings for each treatment?

-L87: depth of soil collected?

-L92: briefly explain why the soil was mixed with perlite

-L96: is 20-28°C considered normal temperature conditions for the bamboo growing season?

-L99: names for treatments should be more logical, i.e., T0, T0.5, T1, T2, T4, T8

-L102: randomly selected?

-L104-108: very unclear; how many pots, and what are the groups? Randomly selected? Successive temperature treatments? What is the “low-temperature treatment” being referred to? Which measurements were taken and when?

-The 20-28°C 6-month experiment and corresponding measurements; and the 5, 0, -5°C culture box part of the experiment and corresponding measurements need to be separated into different subsections

-L110-137: please cite the authority(ies) for all your methods

-L139-141: provide much more information; for example, specify how assumptions were tested and if they were met; your type of experimental design, statistical analyses (ANOVA?), significance level, etc.

Results:

-It would be very useful to run the amendment rates as numeric data in analyses, not just as factors

-Entire section: clarify when reported measurements were taken and at which temperature; specify and clearly define what is being referred to by “cold conditions,” “low temperature,” etc.

-All figures and tables: much more information needed in the captions; for example, what are the errors or error bars, what are the letters, when were the measurements taken and at which temperature, sample sizes, etc.; also, define all abbreviations

-Figure 1: provide letters to indicate statistical differences between treatments

-Table 1, footnote: “soil depth”?

-Table 2: provide the equation from L159

-L174,179,180: discussion, not results

-L188: discussion/conclusion, not results

-Figures 2-6: remove abbreviation in the legend and just keep the temperatures; most have low SEs(?) for the small sample sizes – double-check accuracy; provide trendlines for each temperature across the treatments

-Figure 7: remove dual label for 7A and 7B; add a brief description of each photo in the caption and add arrows pointing to the noteworthy features that are referenced

Discussion:

-Most of this section belongs: a) in the introduction section (or is a repeat of what was already stated in the introduction section); b) in the results section (or is repeated from the results section); or c) in the conclusions section

-Avoid using lists of findings from other studies; the discussion section should be used to discuss and explain the results of this study in the context of the larger picture and field of knowledge; also discuss possible biases of this study, including how the soil properties, processing of the soil, and the addition of perlite may have affected the results compared to a field experiment

Conclusions:

-More explanation and synthesis are needed

-Provide recommended range of Si amendment rate

References:

-Use align left, not justify

Author Response

Dear Sir,

Thank you very much for handling the review of our manuscript (forests-474886) entitled “Effect of soil Si amendment on Phyllostachys praecox response to low temperature in a pot experiment”. We appreciate the insightful comments and suggestions of the both anonymous reviewers as well as the subject editor, and have carefully considered each point brought up.

Based on the comments we received, careful modifications have been made to the original manuscript. Below you will find our point-by-point responses to the reviewer’s comments/questions.

Please note that: Reviewer’s comments/questions are in black. Our responses are in red.

Sincerely yours,

Shunyao Zhuang, Ph.D.

Reviewer 2 Report

General comments:
The study provides sound evidence, which supports the use of Si amendment on Phyllostachys praecox in low temperatures, particularly in the green house. This is a very interesting and useful study but some moderate to significant revision is required to make the paper stronger. Overall, the introduction could do more to provide relevant information on Si-mediated alleviation of freeze stress as well as background information on the physiological indicators used in the study (e.g. their usefulness and what changes in those indicators mean). Other sections of the paper could be improved as well. All reference numbers in the text should be carefully checked with the reference numbers in the Reference section. A number of the in-text reference numbers do not match with the numbers in the Reference section.

Specific comments:

Title
Silicon should be spelled out fully in the title

Abstract
L15. g/kg. Is it per kilogram of soil? Clarify.
L15. Using physiological indicators and physiological responses as you have done here is confusing. You can simply drop physiological indicators from the abstract since the term physiological responses can broadly cover all the analysis you did on bamboo physiology. Subsequently, when you mention first mention physiological indicators in the Introduction, list those parameters, which constitute your physiological indicators so readers can easily follow what you are saying.
L18. Provide the actual p value, e.g. p = 0.01, not just p<0.05. In cases where p is given as 0 or very small (e.g. 10^-6) then you can indicate it as p <0.01; depending on the number of decimal places you are using.
L19-20. A slight decline in biomass compared to the control? Clarify. Was that decline significant?
L21-22: Why is it useful to report that Si content in the leaves was the highest? You should make that clear.
L22-23. Highest Si content of 68.95 mg/kg; is that total Si content for the whole plant or some specific part? Please clarify.
Line 23-24: “With the application of SI, photosynthesis increased”. Do you mean increasing photosynthesis with increasing rate of Si amendment? Clarify. If so, why the reduced biomass at the highest application rate?
L24-26: What does the changes in SOD, POD, CAT, CMP, and MDA activities with Si amendment practically mean?
L26. 0 degrees C is also low temperature. Therefore, do not designate any one of your temperatures as the low temperature. You can simply say, “A low temperature of -5 degrees….”
L28-30. I think your results and regressions rather show that Si amendment is beneficial to a certain point? At the highest Si amendment, there was no increase in biomass; it rather declined slightly. You should make this clear.

Introduction
L43. Yeo et al. is indicated with reference number 15. In the reference list, number 16 is rather Yeo et al. Correct this. There are a number of these mistakes in the paper. Seems like your references numbers in the text are increased by 1 in the reference list. Recheck all your references.
L45. Full name of MDA should first be given in the text and abbreviations used thereafter.
L66-67. Wrong in-text referencing style used.
L53. Change “Despite of…” to “Despite the..”

L1-L58: The introduction should spend more time providing available information on Si-mediated alleviation of freeze stress. As it stands now, the Intro provides much information on other stresses not directly related to this paper and spend little time on Si-mediated freeze stress which is more relevant for this paper.
L60. Change “local economics” to “the local economy”.
L65-66. What does the reported range in Si concentrations signify? Why is it important to mention them here?
L71. Before using the term “physiological indicator” in your objectives/aims, you should clearly state what your specific physiological indicators are early on in the Introduction. Your Introduction should also provide some background information and relevant information about these physiological indicators; what they mean and why we should care about them.

Methods:
L74-81. Do not give general information about P. Praecox, e.g. where it is normally cultivated. Simply talk about the provenance or area where you obtained your bamboo species and general environmental conditions of that area, and the management regimes employed there. This section should be very concise.
L81-82. This was a greenhouse experiment, not a field study. Thus, your study site is the greenhouse not where you obtained the bamboo rhizomes. Reword.
L85. Use Rhizome sprouts instead of rhizome seedlings. (Seedlings typically come from seeds).
L94. Delete “batch” from “bamboo batch pot experiment”. Change “at 2016” to “in 2016”.
L96. Be consistent; delete basin and stick with pots.
L100-101. Delete “During experimentation, the ”. List those photosynthesis parameters.
L103. You describe how Si content in plants was measured below. Therefore, delete this from here. How was soil available Si measured? How was pH measured; with what instrument?
L104-108. This part should be written more clearly. 1). Let it be clear from the beginning that, the remaining pots of each treatment, after the destructive sampling, were subjected to 3 temperature treatments, with five replicates for each temperature treatments.2). What are functional leaves? Clarify. 3). Low-temperature treatment is quiet ambiguous. It could be taken to mean only the -5 degrees C treatment. Therefore use a different term, example, Low-temperature treatments.
L113-114. Give a brief and concise description of how Si molybdenum blue method works or at least provide a reference for that procedure.
L121: More relevant information is needed here. How many leaves were measured for each bamboo plant? How many bamboo plants were measured in a single pot? How many pots per treatment were measured?
L123-124. At the least, provide reference(s) for those procedures.
L127. Why only T0 and T4? What about the other treatments, why were they not included?
L139. What are those statistical indices? Indicate them. What type of regression? Linear, multiple, non-linear? Clarify. Change “formulae” to coefficients.
L140. Why did you choose Duncan's Test, and not and not any of the other multiple comparison statistical test? Delete “by”.

Results
L144. Provide p values through out the text, for any place where you indicate a significant increase/decrease.
Fig. 1. (1). What is biomass increase ratio? You mean biomass increment in percentage? Might be easier to simply use biomass increase rather than biomass increase ratio. The term "biomass increase ratio" was not indicated anywhere in the methods, and therefore should not just appear in the results. (2). You make use of root-canopy ratio (%) in the results, but you did not indicate that in the Data analysis. How was that calculated? Why did you choose that index, and what does it mean?
L. 155. Si content greatest in leaves. What is the importance of this? Expand on this in the Discussion.
L.157. Do you mean “bamboo items”?. What does that refer to? Reword through out the text.
L159. The form of the equation should have been indicated in the data analysis. Simply stating that a quadratic function was used to describe the relationship between Si content in plant and rate of Si amendment would suffice.
L.162. I disagree with this conclusion. Your quadratic function (with the negative coefficient for the quadratic term) suggests that with increase in Si amendment, there is increase in uptake until maximum uptake is achieved at some increased Si level after which additional increase in Si amendment may result in a decline in uptake. Your equations do not say anything about maximum uptake regardless of Si amendment rate.
L169. Provide p-values for the regression coefficients. Or you can rather show in bold font those coefficients that are not significant.
L170-175. Expand on the practical importance of these results in the Discussion. Provide p-values where you indicate significant increase/decrease.
L179-181. Change “By Si amendment” to “With Si amendment…”. Also, was the reported increase significant? Provide p values.
L182. Greatly reduced. Was that a significant decrease? Provide p-values.
L. 184-185. Is this increase significant? Provide p value.
L.186-188. Are the reported increase/decreases significant?
L 177-188. For reporting increasing/decreasing trends of physiological indicators with Si amendment, you should also provide a regression of these indicators with Si amendments. The regressions are important to determine if these trends are significant or not by providing us with p-value for the regression coefficients.
L202. Change “grana suffered from disintegration” to “the grana disintegrated”.

Discussion
L. 216. What is bamboo growth area?
L. 218-219. “Si favors…. Accordingly, Si… in P. praecox”. Combine these sentences.
L.221. What is a reasonable amendment rate? You should give a range or mean value from your study.
L225-228. The sentence is too long. Divide into 2 or 3 separate sentences.
L223-231: Not very clear which result you are describing. Make such linkages very clear through out the Discussion.
L234-240. Make sure all the abbreviations you are mentioning for the first time have been written out in full. Do not cite your Figures in the Discussion.
L.250. Wrong in-text referencing style.
L254-260; Are you reporting what you found in your study or just reporting what is already available in the literature? It seems like you are drawing on the literature to show how Si improves bamboo cold tolerance rather than drawing attention to how your result show the mechanism of Si in improving bamboo cold tolerance, which is more interesting.

Conclusion

This is a green house study, and it must be noted that care must be taken in extrapolating findings here to field situations. Field test are needed.

Author Response

(The authors gave the same response as above.)

Round 2

Reviewer 1 Report

The authors have addressed many of my concerns and have improved the manuscript. However, there remains a lack of clarity regarding the measurements and the results that needs to be addressed (specifically, further clarity related to the 6-month pot experiment and the low temperature incubation). In general, more attention to detail is needed to improve the overall quality of the manuscript and the presentation. More specific comments are below.

1.     The introduction is often overpacked with information without clear connection to the study and coherency of thought

2.     The authorities for many methods used in this experiment are not properly cited

3.     Section 2.2.1. – the non-random selection of plants for the low temperature incubation is concerning because it may have biased results

4.     Section 2.5. can be combined with Section 2.4. as long as it is clear which plants after which treatments underwent which measurements

5.     The data analysis section is unclear and difficult to follow in places

6.     No need to provide the alpha level after providing the actual p-value in parentheses

7.     No need to continually repeat the application rates of treatments after the treatments have been defined

8.     Figure 1 – add letters representing significant or non-significant differences for the root-canopy ratio

9.     All figures and tables – include information provided in footnotes in the caption instead of in footnotes

10.  Figure 7 – It seems most arrows and identifying letters would be more visible in white; also make them thicker/bold (i.e., as big and visible as possible without obscuring important parts of the figure

11.  The discussion is much improved, but is still a bit of a mix of introduction information and results; work on creating connections and organization

12.  It seems the recommendation should be at least 2 but less than 8 g/kg of Si fertilization for use for verification of the results of this study under field conditions

13.  The 8 g/kg treatment did not cause a decline in biomass compared to the control, which the wording implies at multiple points in the manuscript; clarify

Author Response

Dear Ms. Wei,

Thank you very much for handling the review of our manuscript (forests-474886). We appreciate the insightful comments and suggestions of anonymous reviewers. Based on the comments we received secondly, careful modifications have been made to the revised manuscript. Below you will find our point-by-point responses to the reviewer’s comments/questions.

Please note that: Reviewer’s comments/questions are in black. Our responses are in red.

Sincerely yours,

Shunyao Zhuang, Ph.D.
